# Architecture of the Heme-translocating CcmABCD/E complex required for Cytochrome *c* maturation

Lorena Ilcu[1], Lukas Denkhaus[1], Anton Brausemann[1], Lin Zhang [1]✉ & Oliver Einsle [1]✉

Mono- and multiheme cytochromes *c* are post-translationally matured by the covalent attachment of heme. For this, *Escherichia coli* employs the most complex type of maturation machineries, the Ccm-system (for <u>c</u>ytochrome <u>c</u> <u>m</u>aturation). It consists of two membrane protein complexes, one of which shuttles heme across the membrane to a mobile chaperone that then delivers the cofactor to the second complex, an apoprotein:heme lyase, for covalent attachment. Here we report cryo-electron microscopic structures of the heme translocation complex CcmABCD from *E. coli*, alone and bound to the heme chaperone CcmE. CcmABCD forms a heterooctameric complex centered around the ABC transporter CcmAB that does not by itself transport heme. Our data suggest that the complex flops a heme group from the inner to the outer leaflet at its CcmBC interfaces, driven by ATP hydrolysis at CcmA. A conserved heme-handling motif (WxWD) at the periplasmic side of CcmC rotates the heme by 90° for covalent attachment to the heme chaperone CcmE that we find interacting exclusively with the CcmB subunit.

Cytochromes *c* are a diverse and versatile family of metalloproteins present in all kingdoms of life[1]. They contain one or multiple copies of the tetrapyrrole cofactor Fe-protoporphyrin IX, or heme, and their defining characteristic is the covalent attachment of this metalloorganic moiety to characteristic binding motifs of the sequence C-X$_n$-C-H (most commonly $n = 2$; Supplementary Fig. 1a)[2]. The purpose of this post-translational modification is at least twofold. First, cytochromes *c* are invariably deployed in an extracytoplasmatic environment, so that the covalent attachment serves to prevent cofactor loss. Second, fixing the heme groups to the peptide chain alleviates the need to form bulky binding pockets, allowing for a very high cofactor:protein ratio and with it the formation of tightly packed chains of heme groups within a single protein[3]. Consequently, the typical roles of cytochromes *c* are in electron transfer or in catalyzing multi-electron redox reactions[4]. The most prominent member of this protein family arguably is the monoheme cytochrome *c* from the mitochondrial respiratory chain, where it shuttles electrons from the cytochrome $bc_1$ complex to

cytochrome *c* oxidase[5]. However, prokaryotes have evolved systems of far higher complexity[6], with several and up to dozens of heme groups on a single peptide chain. Select organisms such as *Geobacter sulfurreducens* encode more than one hundred multiheme cytochromes *c* in their genome[7]. The maturation of cytochromes *c* is elaborate. Apocytochromes contain a signal sequence for export from the cytoplasm via the Sec translocon, and the nascent peptide chain is then processed in the extracytoplasmatic compartment by a heme lyase complex that recognizes the heme-binding motifs and mediates the covalent attachment of the cofactor by adding the two cysteine thiol groups to its vinyl side chains. Most remarkably, this process is independent of the overall sequence of the protein and works for apocytochromes of vastly different size. The peptide chain is scanned by the lyase that specifically recognizes heme-binding motifs. In all known heme lyases, proteins of the 'heme-handling' family play a crucial role, and their hallmark is a tryptophan-rich motif, shorthanded as 'WxWD motif', by which the cofactors are held and positioned for interaction with the

[1]Institut für Biochemie, Albert-Ludwigs-Universität Freiburg, 79104 Freiburg im Breisgau, Germany. ✉e-mail: lin.zhang@biochemie.uni-freiburg.de; einsle@biochemie.uni-freiburg.de

protein[8]. As cytochromes *c* are an evolutionarily old protein family the maturases have diversified over time, and current classifications list five different systems of heme lyases[9,10]. The most complex of these is system I, the Ccm system (for <u>c</u>ytochrome <u>c</u> <u>m</u>aturation) that is found in α- and β-proteobacteria, plant mitochondria, deinococci, archaea and some γ- and δ-proteobacteria. In most cases it is encoded in a single operon of the composition *ccmABCDEFGHI* (Supplementary Fig. 1b), whose protein products form two membrane-integral protein complexes with distinct functionality[11]. Here, a CcmFHI complex serves as the heme lyase module that scans the apopeptide, reduces disulfides that may have formed in the oxidizing environment via a thioredoxin module and attaches heme cofactors at the core lyase subunit CcmF[12,13]. CcmG is a DsbA-like thioredoxin that provides reducing equivalents to the lyase complex. The second module is CcmABCD, another integral membrane protein complex centered on the ABC transporter CcmAB. It was described as a heme exporter that translocates the cofactors across the membrane to attach them to the heme chaperone CcmE in the periplasm[14]. CcmE is a monotopic membrane protein that folds into a compact β-barrel domain and forms a covalent link to heme with a histidine residue located in its flexible C-terminal tail[15,16]. CcmE is the element that connects the heme translocation complex CcmABCD and the heme lyase complex CcmFHI[17], and although CcmF invariably contains a *b*-type heme group, it is only cofactors delivered via CcmE that are linked to the apo-cytochrome chains (Supplementary Fig. 1c).

We have recently reported the structure of a core subunit of the heme lyase, the CcmF protein from *Thermus thermophilus*, determined by X-ray crystallography[18]. The structure revealed an unprecedented fold for membrane proteins and showed the accessory *b*-type heme group located at the bottom of a funnel-like channel that seems to span the entire membrane. However, while the *b*-type heme was located within the cytoplasmic leaflet, the periplasmic side of the channel had a wide lateral opening into the membrane. Based on the size and shape of this entrance and the adjacent protein cavity we proposed a 'buoy model' for the action of CcmE, where the chaperone does not actually extract the heme group into the periplasm, but rather retains the hydrophobic cofactor within the lipid bilayer to guide it into the opening in CcmF[18]. To clarify how CcmE works in conjunction with the lyase, we proceeded to investigate the other key interaction of the chaperone, its loading with heme at the CcmABCD complex. In this work, we produce and isolate CcmABCD and CcmABCDE complexes from the γ-proteobacterium *Escherichia coli* and determine their structure by cryo-electron microscopy (cryo-EM) single particle analysis. Most recently, Kranz, Zhang, Zhu and co-workers also reported structures of a Ccm(AB)₂CD complex from *E. coli*[19]. We instead find a predominant stoichiometry with two copies of CcmCD and identify the binding mode of the chaperone CcmE to this complex. Our data further support the buoy model of CcmE action and suggest that this unusual ABC transporter acts as a heme floppase.

## Results

*E. coli* utilizes a *ccmABCDEFGH* operon for cytochrome *c* maturation, in which the CcmH protein represents a fusion of the CcmH and CcmI proteins found separately in other organisms that employ system I[20]. The model prokaryote encodes only six *c*-type cytochromes in its genome that are exclusively produced during anaerobic growth. To allow for the recombinant production of *c*-type cytochromes under oxic conditions, Thöny-Meyer and co-workers have created the accessory plasmid pEC86 that expresses the Ccm system from a constitutive promoter in trans[21]. Here we generated a series of expression constructs for components of the Ccm system of *E. coli* and isolated a CcmABCDE complex that was solubilized with DDM or GDN and isolated by affinity and size exclusion chromatography (Supplementary Fig. 2a, b). This sample was then characterized by cryo-EM single-particle analysis, and classes representing a Ccm(ABCD)₂ and a

Ccm(ABCD)₂E complex (see below) were refined, leading to structural models at resolutions of 3.47 Å and 3.81 Å (Supplementary Fig. 3).

## Structure of the CcmABCD complex

The CcmA and CcmB proteins are the respective ABC modules and transmembrane subunits of a canonical ABC transporter. They were originally suggested to act as a heme exporter[14], assembling a Ccm(AB)₂ heterotetramer according to the canonical architectural principles of ABC transporters. The tetramer then was found to form a complex with the heme-handling protein CcmC that contains the WxWD motif[17], and the small but essential monotopic CcmD[22]. In line with this, the recent cryo-EM analysis reported a Ccm(AB)₂CD complex[19] that was previously quite accurately predicted by AF2Complex[23], which in turn builds on AlphaFold[24]. In our analysis, the Ccm(AB)₂ unit is also at the core of the complex, but already in initial 2D classes it immediately became apparent that the assembly was symmetric (Fig. 1a), with CcmCD modules on both sides of the core transporter, resulting in a Ccm(ABCD)₂ heterooctamer with a total mass of 164 kDa and 26 transmembrane helices (Fig. 1b, c; Supplementary Fig. 4). The ABC transporter was in a nucleotide-free, inward-open state, without direct contact between the two nucleotide-binding CcmA subunits. The 201 aa CcmA attained the canonical ABC domain fold related to P-loop ATPases, with a P-loop (Walker A motif) [36]G<u>SNGA</u>GKT[43] that forms the nucleotide-binding site, as well as a *switch I* region [179]GIVLTTHQP[188] and a *switch II* region (Walker B motif) [152]L<u>DEPF</u>TAIDV[161] that typically undergo conformational changes upon nucleotide binding, and the lid loop [81]IGHQPGIKTRL[91] that is crucial for interaction with CcmB (Fig. 1d). In all motifs, the conserved residues given in bold are involved in binding MgATP. In the nucleotide-free state of the complex there were no direct interactions between the two copies of CcmA, their centers of mass were 36 Å apart.

CcmB is an integral membrane protein with six transmembrane α-helices and a characteristic, amphipathic "elbow helix" at its N-terminus, situated parallel to the membrane plane and held in place by ionic interactions between the conserved D73 and K96 as well as E10 and R68 (Fig. 1e). Connecting helices *h*II and *h*III on the cytoplasmic side of the membrane, CcmB contains the characteristic coupling helix from residues 78-84 that mediates the interaction of the transmembrane subunit with the nucleotide-binding subunit CcmA[25]. Phylogenetically, CcmAB groups with the exporter family ABC-A2 of ABC transporters[26] or—according to a more recent classification—with type V exporters (floppases)[27]. System I cytochrome *c* maturases are also found in plant mitochondria, and in *Arabidopsis thaliana* the gene for the ATPase CCMA is encoded in the nuclear DNA, while *ccmB* is found in the mitochondrial genome[28]. The CcmC protein is also an integral membrane protein with six transmembrane helices (Fig. 1c). Together with the heme lyase components CcmF and Ccs(B)A it constitutes the family of 'heme-handling proteins', with a [114]W<u>X</u>KXX<u>W</u>GXΩ<u>W</u>X<u>WD</u>XRLT[130] motif (CcmC numbering), shorthanded 'WxWD', in the loop region connecting transmembrane helices *h*III and *h*IV on the periplasmic side of the membrane (Fig. 1c, f)[8]. The terminal aspartate residue of this motif, D126, caps helix IV by accepting short hydrogen bonds from the backbone amide nitrogen atoms of residues A127 and R128 (see below), providing a negative charge to compensate for the dipole moment of the helix in an arrangement very similar to the ones observed in the crystal structure of CcmF of *T. thermophilus*[18] and the cryo-EM structure of CcsBA[29] (Supplementary Fig. 5). The final subunit of the complex, CcmD, was reported to be required for the dissociation of CcmE after transfer of the heme group[22,30]. CcmD has its N-terminus in the periplasm where it begins with an amphipathic helix that places the first 18 residues parallel to the membrane boundary (Fig. 1g). From residue A18[D] to M41[D], it traverses the membrane exclusively with hydrophobic residues, with the notable exception of H38[D] that resides within the membrane, about 5 Å from

the cytoplasmic face. Exiting the membrane, CcmD extends more than 35 Å into the cytoplasm. As indicated by the increasing disorder of the cryo-EM maps (Fig. 1g), the C-terminus of CcmD is rather flexible, and we never observe a direct interaction with the nearby CcmA subunits. However, residue R54[D] forms three short hydrogen bonds to the short helix at the C-terminus of CcmC that also orients parallel to the cytoplasmic face of the membrane. The CcmABCD/E samples co-purified with bound heme *b* in an oxidized state, as indicated by an α-band at 560 nm upon reduction (Supplementary Fig. 2c). No bound heme was found in the structural analyses of different complexes, while heme-bound structures were obtained in the recent report on a Ccm(AB)₂CD complex, but here external hemin was added prior to cryo-EM analysis[19]. We had not followed this strategy, and this presumably led to the loss of the non-covalently bound cofactor during grid preparation.

The overall architecture of the CcmABCD complex in its nucleotide-free state thus is a C2-symmetric heterooctamer with dimensions of 100 Å × 80 Å × 60 Å. It contains a pronounced groove between the CcmB, CcmC and CcmD subunits on either side that is capped by the WxWD loop of CcmC at the periplasmic interface. For the following discussion, we designate this face of the complex the 'front' and the opposite side, with a rather flat CcmBC interface, its 'back' (Fig. 1b).

## Conformational states of CcmAB

To obtain structural information on the ATP-bound state of CcmAB, we generated a E154Q[A] variant that replaced the catalytic glutamate in the *switch II* region of the nucleotide-binding subunit (Fig. 1d) to obtain a protein that is incapable of ATP hydrolysis but still binds the nucleotide[31,32]. This variant was isolated as the wild type

(Supplementary Fig. 2a, b), but did not contain bound heme (Supplementary Fig. 2d). As anticipated, the single point mutation in the nucleotide-binding site of CcmA locked the transporter in a closed conformation, shortening the distance between the centers of mass for the two CcmA subunits from 36 Å to 29.4 Å (Fig. 2a). No heme was bound in this sample, and we obtained one data set that contained CcmA without and one with bound ATP, with only minor changes to the overall conformation of the complex (Supplementary Table 1). From the inward-open state of the nucleotide-free Ccm(AB)₂ heterotetramer, the E154Q[A] variant switched to a fully closed configuration with no internal cavity of sufficient size to accommodate a cargo molecule of the size of a heme cofactor. The binding sites for ATP were located close to the interface of the CcmA dimer (Fig. 2b), with residue S130 from the other monomer involved in the coordination of the ATP phosphates. ATP bound to its canonical position in this class of proteins, with the P-loop cradling the triphosphate near the *switch I* and *switch II* regions and the lid loop (Fig. 2c). The overall conformational changes in the different states of CcmA were subtle, and the nucleotide-free wild-type structure (Fig. 2d) showed the same conformation of the P-loop as the nucleotide-free E154Q[A] variant (Fig. 2e), although the latter switched the transporter into its closed state. Only upon binding of MgATP the conformation of the P-loop changed substantially, including in the lid loop where H83 coordinated the Mg²⁺ cation (Fig. 2f).

Unexpectedly, the overall stoichiometry of the CcmABCD changed for the E154Q[A] variant, in that the closure of the core ABC transporter Ccm(AB)₂ consistently led to the loss of one copy of CcmCD, resulting in a Ccm(E154QAB)₂CD complex (Fig. 3a, Supplementary Fig 6, Supplementary Movie 1). This fundamental change in quaternary structure led to a structural asymmetry of the two CcmB protomers.

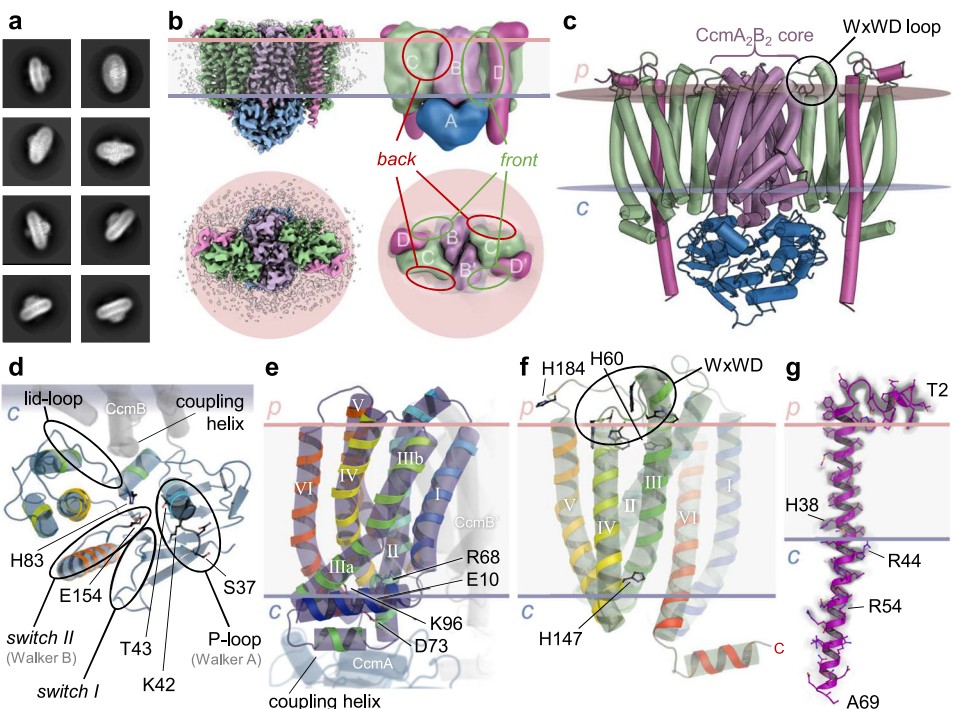

**Fig. 1 | Architecture of the heterooctameric Ccm(ABCD)₂ complex. a** 2D classes of the CcmABCD data set showed a symmetric envelope in an elliptical micelle. **b** Single-particle reconstruction of nucleotide-free Ccm(ABCD)₂ at 3.47 Å resolution in front (above) and top view from the periplasm (below). In the schematic representation on the right, the subunits, as well as the front and back faces of the CcmBC interface are labeled. **c** Cartoon representation of the 194 kDa complex in the membrane. Subunit coloring according to (**b**). **d** The ATPase subunit CcmA in the nucleotide-free state, colored from blue at the N-terminus to red at the

C-terminus. Conserved motifs of P-loop ATPases are indicated, as well as the residues discussed in the text. **e** CcmB, the transmembrane subunit of the core ABC transporter. TM helices are labeled and the position of the membrane and of the adjacent CcmA and CcmB' subunits are indicated. **f** The heme-translocating subunit CcmC, with the heme-handling motif WxWD in the loop connecting TM helices *h*III and *h*IV. **g** Subunit CcmD, with a single TM helix, was fully defined and extends 35 Å into the cytoplasm.

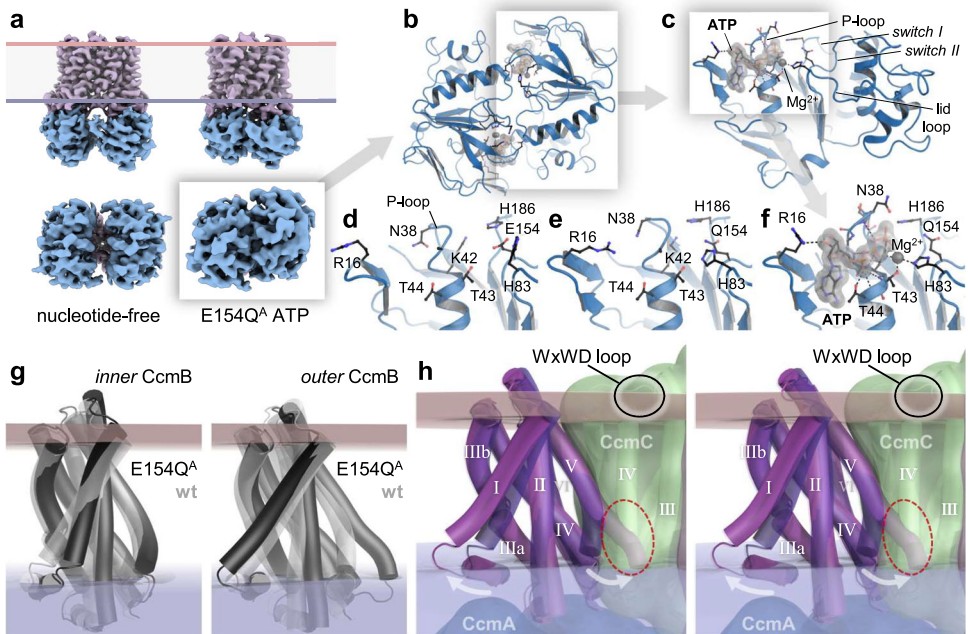

**Fig. 2 | Closed state of CcmAB and structural asymmetry in CcmB. a** Single-particle reconstruction of nucleotide-free CcmAB in the open state (left) and of the ATP-bound E154Q[A] variant in the closed state (right). **b** The CcmA dimer of the E154Q[A] variant with 2 ATP bound at the dimer interface (gray). **c** Detail of the ATP-binding site in one monomer of CcmA with the functional motifs of the P-loop NTPase family. **d** Conformation of the P-loop in native CcmA (open state). **e** Conformation of the P-loop in the E154Q[A] variant without bound ATP. **f** ATP-binding in the E154Q[A] variant. **g** Comparison of the conformations of the *inner* and *outer* CcmB monomer (Ref. Fig. 3a) in the nucleotide-free wild type (gray) and the E154Q[A] variant (black). In the outer monomer, the substantial switch of helix *h*V only occurs in the asymmetric ATP-bound complex. **h** Stereo image of the asymmetric conformational change observed in the unobstructed monomer of CcmB (transparent), superimposed on the second monomer that remained bound to CcmC (solid). The extension of *h*V of CcmB would result in a steric clash with helix *h*IV of CcmC.

The inner protomer of CcmB (Fig. 3a) largely retained its open-state conformation, as was the case for the adjacent, remaining copies of CcmC and CcmD. The outer protomer, however, now unrestrained by interactions with CcmCD, underwent a conformational change that most strongly affected the cytoplasmic termini of its transmembrane helices *h*I and *h*V (Fig. 2g). In the outer protomer of the E154Q[A] variant complex, the cytoplasmic ends of both helices moved outward from the core of the CcmB dimer, by 7.0 Å in the case of *h*I and 8.1 Å for *h*V (Fig. 2g, h). This latter shift would have caused a significant clash with transmembrane helix *h*IV of CcmC and is therefore the likely reason for the dissociation of the second copy of CcmCD from a heterooctameric Ccm(ABCD)₂ complex (Supplementary Movie 2). The altered subunit stoichiometry now also corresponded to that of the recently reported Ccm(AB)₂CD structures, where AMPPNP- (PDB 7F03) and ATP-bound (PDB 7F04) structures showed a closed conformation similar to that of our E154Q[A] variant, while a second 3D reconstruction from the ATP-bound sample with bound heme was designated as semi-open (PDB 7VFP), with a conformation very similar to the nucleotide-free state determined here[19]. Li et al. also reported a nucleotide-free open state with a larger distance between the two copies of CcmA (PDB 7VFJ). We did not find this form, but we hypothesize that this wider opening of the transporter may only be possible due to the absence of a second CcmCD module in their preparations. Although it was not discussed by the authors, these structures also showed a consistent asymmetry of the two copies of CcmB that is least pronounced in the wide open apo form of the complex (PDB 7VFJ). Only here, helices *h*I and *h*V point inward, but in the absence of a second copy of CcmCD helix *h*I is still less buried within the CcmB dimer than in our structure of the symmetric Ccm(ABCD)₂ complex. In all other structures, helices *h*I and *h*V pointed outward in the unrestrained, outer protomer only, which therefore may also have contributed to the release of a second copy of CcmCD that gave rise to the observed, asymmetric complexes.

## CcmC and CcmD and translocation of heme

At the extracellular side of the membrane, the difference between the open and closed states of CcmAB was minimal, and neither configuration showed an 'outward open' state or featured an apparent binding site for a cargo molecule. In contrast, the observed conformational change of the outer protomer of CcmB in the ATP-bound state or the E154Q[A] variant complex (Fig. 3a) implies that an adjacent CcmC protomer would have to either dissociate from the complex or change its own conformation in response (Fig. 2h, Supplementary Movie 2). In CcmC, the heme-handling motif [119]WGT**W**WV**W**D[126] forms the loop connecting helices *h*III and *h*IV (Fig. 1f). In our structures of CcmC, and also in those published recently[19], these helices were packed close together and the WxWD loop pointed towards the front face of the complex. In the reported structures of a Ccm(AB)₂CD complex with bound heme (PDB 7F04 and 7VFP) the position of the helices was similar, but the heme cofactors were bound to CcmC at the back side of the complex (Fig. 3b). Overall, the heme-bound structures 7F04 and 7VFP align to the symmetric Ccm(ABCD)₂ complex reported here with a root-mean-squared displacement of only 3.8 Å for all atoms, but the conformations of the individual subunits are more similar, with only 0.6 Å r.m.s.d. for CcmC, irrespective of the presence of a bound heme. The WxWD loop is not in contact with the heme group and remains almost completely unchanged. The same was observed for loop 5, the partially disordered, extended protein stretch connecting helices *h*V and *h*VI of CcmC (Fig. 3c). While this loop was not fully resolved in either structure, the positions of the two histidines of CcmC that bind heme, H60 and H184, are largely identical. In 7F04, only H60 coordinates the heme iron, while H184 points towards CcmB (Fig. 3c).

We then proceeded to conduct various structure predictions of the CcmABCD complex with AlphaFold2 in multimer mode[24]. The predicted structures were very well in line with the experimental ones

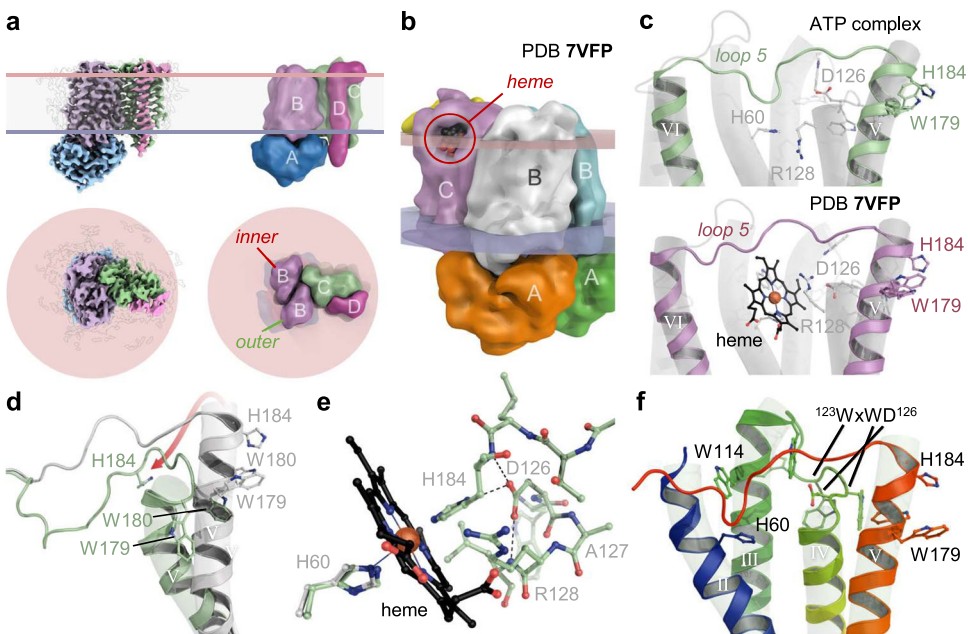

**Fig. 3 | Role of the CcmBC interface in heme translocation. a** Single-particle reconstruction and schematic representation of The Ccm(AB)$_2$CD complex isolated for the E154Q$^A$ variant in front (above) and top view (below). **b** Heme bound to CcmC in a structure of a Ccm(AB)$_2$CD complex from *E. coli* (PDB 7VFP). Subunit coloring according to the original publication. **c** Arrangement of the flexible loop 5 in CcmC without (above) and with bound heme (below, PDB 7VFP). **d** Reorientation of helix *h*V and loop 5 in the AlphaFold2 prediction (green) relative to the experimental structure with bound ATP (white). **e** Superposition of the AlphaFold2 prediction (green) with heme as bound in PDB 7VFP to H60 only (black). In the *in-silico* prediction, the position of H184 makes it a suitable distal axial ligand to the heme iron. **f** The four-helix bundle motif around the WxWD loop in CcmC in the ATP complex structure, colored from N- to C-terminus in a rainbow color ramp.

but exhibited several, potentially relevant differences. Individual AlphaFold2 runs produced almost identical predictions for individual subunits, so that this analysis focuses on the largest assembly, a symmetric Ccm(ABCDE)$_2$ complex (Fig. 3d). The model had the core ABC transporter Ccm(AB)$_2$ in a closed state, very similar to the ATP-bound E154Q$^A$ variant complex. However, in the symmetric assembly the helices *h*I and *h*V of the CcmB subunits did not project outward, so that the AlphaFold2 model did not reproduce this key feature of the closed-state experimental structures. The predicted CcmC subunit differed most strongly from the one in any of the experimental structures (Supplementary Fig. 5), and beside a displacement of the WxWD loop by approximately 1.5 Å this difference was primarily due to a major tilt of the periplasmic half of helix *h*V of CcmC (Fig. 3d). With this tilt, the extended loop 5 folded towards CcmC, leading to a shift of H184 by 18 Å, away from its position close to CcmB in the experimental structures (Fig. 3e). Although AlphaFold2 predictions do not currently include cofactors, the two histidines previously identified to act as axial ligands to heme, H60 and H184, were directly juxtaposed and would be well suited to jointly ligate heme that is bound in a similar mode as in structure 7F04 (Fig. 3f)[23]. An AlphaFold2 prediction of only the CcmC subunit showed the exact same conformation, underlining that this prediction is not based on the formation of a complex with CcmB. Furthermore, H184 now formed a hydrogen bond from its N$_{\delta1}$ atom to the β-carboxylate of D126 of the WxWD motif (Fig. 3e). The AlphaFold prediction for CcmC can also be compared directly to the experimental structure of an open conformation of the system II heme lyase CcsBA from *Helicobacter hepaticus*[29]. In this structure (PDB 7S9Y), heme binds at the periplasmic side of the integral membrane protein, with H761 and H897 as axial ligands (Supplementary Fig. 5e). As noted previously[33,34], the family of heme-handling proteins shares a common folding core, and helices *h*II-*h*V of CcmE align well with helices *h*IX-*h*XII of the 13-TM-helix CcsBA lyase[29], and with helices *h*V-*h*VIII of CcmF (Supplementary Fig. 5c)[18]. The central part of this consecutive four-helix bundle, loop 2 (connecting the second and third helix) on the periplasmic side of the membrane is the one containing

the WxWD motif, while the axial histidine ligands for the heme cofactor reside in the first helix and in loop 4, respectively. The domain opens to the periplasmic side, and here the bound heme group is cradled in the structures of CcsBA (PDB 7S9Y) and CcmC (PDB 7F04). Note that in CcsBA heme is presumably transported through a channel formed by the N-terminal helices of the protein (a separate CcsB protein in other organisms), and that in CcmF heme delivery from the chaperone CcmE likely occurs via a lateral gate in the C-terminal part of CcmF[18]. In either case the cofactor arrives at the other side of the WxWD motif that might then act to grab the porphyrin ring through aromatic stacking with its tryptophan residues. All this is in line with a consistent function of the WxWD motif in rotating the heme group by approximately 90° and moving it into the holding position observed in the two structures discussed above, such that its 8-vinyl group points towards the periplasm. As the lyase forms the covalent link to the cofactor, this 8-vinyl group is attacked first by the second cysteine of a heme-binding motif in the apocytochrome, while CcmE in system I is linked to heme via its 3-vinyl group[35,36]. For the present analysis of the Ccm(ABCD)$_2$ complex, this suggests that the heme in structure 7F04 is also observed in a binding pocket that it reaches after being passed *through* the WxWD motif. For this, the helices *h*III and *h*IV of CcmC that this motif connects are too close in all known structure of the heme-handling core domain, implying a conformational rearrangement during heme handling (see Discussion).

**Interaction with the heme chaperone CcmE**
The construct used to produce the *E. coli* CcmABCD complex also contained the *ccmE* gene encoding the heme chaperone CcmE that receives the cofactor from the CcmABCD complex and shuttles it to the heme lyase complex CcmFH(I)[37,38]. CcmE is a monotopic membrane protein with a periplasmic β-barrel domain that has been previously characterized by NMR spectroscopy[16,39] and binds the heme group it receives from the CcmABCD complex covalently via the conserved H130 at its C-terminus, while residue Y134 was suggested to serve as an axial ligand[15,38]. In our cryo-EM single particle data of the

native complex, the two most highly resolved 3D classes contained the Ccm(ABCD)$_2$ complex (Supplementary Fig. 3), but one of these additionally featured a prominent periplasmic domain that after refinement was modeled as CcmE with its N-terminal transmembrane helix (Fig. 4a). This helix interacted exclusively with helix *h*VI of CcmB, with the periplasmic domain of CcmE poised above the CcmBC interface on one side of the complex (Fig. 4b, Supplementary Movie 3). Although the observed interface was small, the interaction of CcmB with CcmE was well resolved in the maps. 29 % of all particles used for the final refinement contained CcmE, and although at lower contour levels a second copy of the chaperone on the opposite side of the complex was apparent and the N-terminal helix of CcmE was visible in lower-resolution reconstructions during data processing (Fig. 4c), the relevant class predominantly contained a single copy of CcmE, with slightly lower local resolution (Supplementary Fig. 3) and presumably substoichiometric occupancy. The transmembrane helix of CcmE was modeled from residue N2 to S32 and is followed by a short, flexible hinge preceding the β-barrel domain that commences at F37. The heme-coordinating residues H130 and Y134 faced the membrane but were situated above CcmB rather than CcmC (Fig. 4b). The association of CcmE to the complex also did not seem to affect its structure, and there was no discernible change observed in CcmB.

AlphaFold2 yielded a model of a symmetric Ccm(ABCDE)$_2$ complex (Fig. 4d) with a root-mean-squared deviation from the experimental structure of 3.1 Å for all atoms (Supplementary Fig. 7) that also reproduced the observed association of CcmE to the heterooctameric core very well. In the prediction, two copies of CcmE bound peripherally to CcmB, and the binding mode for the N-terminal helix of CcmE precisely matched the one observed in our cryo-EM analysis. This was

not necessarily expected given the weak and possibly transient interaction of CcmE with CcmC. However, the periplasmic domain of CcmE was rotated by about 90° towards CcmC, placing it above loop 5 and the WxWD motif of the latter (Fig. 4e). In the AlphaFold model, the heme-receiving residue H130$^E$ stacked onto W119$^C$ of the extended WxWD motif (Fig. 4f), in a position very well suited to attack the heme group if bound as seen in PDB 7F04 (Fig. 3e, Supplementary Movie 3). In the experimental structure, the periplasmic domain of CcmE showed enhanced mobility and a lower local resolution than the core complex, which in the absence of a bound heme group is not surprising. According to the role of CcmE in our buoy model of heme transport from the Ccm(ABCD)$_2$ translocase to the CcmFGH(I) lyase, holo-CcmE is not required to attain a rigid conformation, as the heme cofactor is never fully extracted from the membrane[18].

## Discussion

The open and closed conformations of CcmABCD reported here confirm the recent work by Li and co-workers[19], but differ in several aspects of possible functional relevance. Most obviously, we have isolated a heterooctameric Ccm(ABCD)$_2$ complex as a predominant species in the nucleotide-free state, while the structures reported previously consistently only contained one copy each of CcmC and CcmD. The reasons for this difference are not obvious. Both groups cloned a polycistronic, genomic DNA fragment of *E. coli* with an added N-terminal affinity tag at CcmA and solubilized the membrane fraction with 1% of DDM. In our case, a StrepTag(II) was used and cells were disrupted by sonication, while Li and co-workers used a hexa-histidine tag and a microfluidizer for cell lysis. Our centrifugation steps were shorter, and we changed detergent to the synthetic digitonin analog

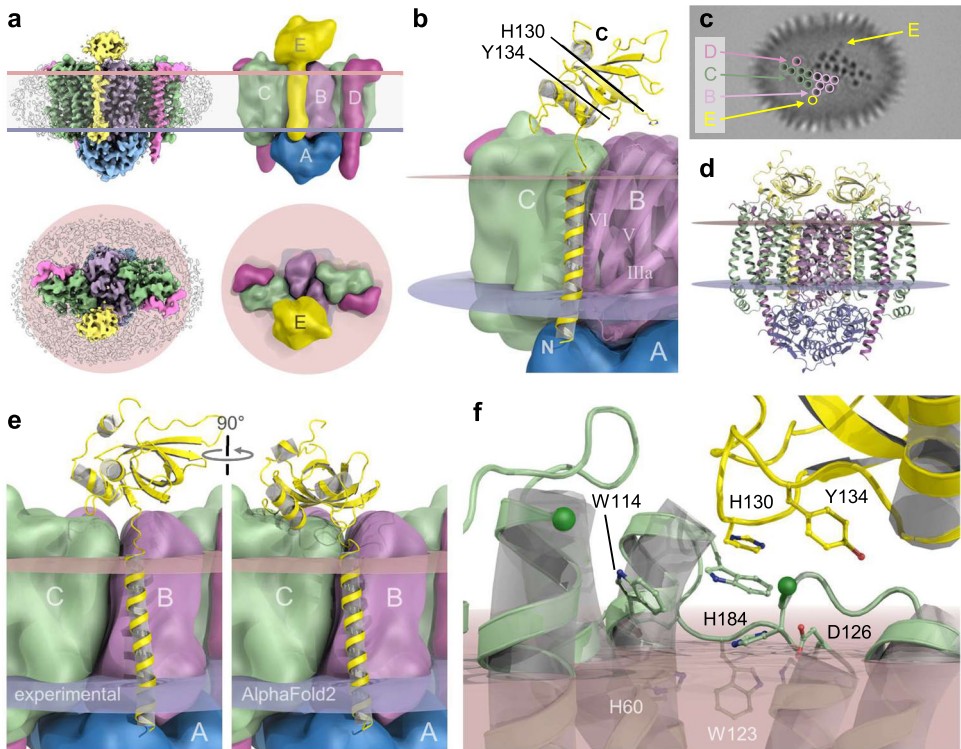

**Fig. 4 | Binding of the heme chaperone CcmE to the Ccm(ABCD)$_2$ complex. a** 3D class from the wild type data set that contained bound CcmE in front (above) and top view (below). **b** Position of CcmE at the back side of the complex, with the TM-helix of CcmE in exclusive contact with helix *h*IV of CcmB. **c** Central slice through a low-resolution (8 Å) 3D class of CcmABCDE, showing the transmembrane helices of the complex and their assignment to subunits. Note that the additional TM helix of CcmE is visible on either side of the complex. **d** AlphaFold2 model of a

Ccm(ABCDE)$_2$ complex, in good agreement with the experimental data (Supplementary Fig. 7). **e** The main difference between the experimental structure and the AlphaFold2 model is in the positioning of the periplasmic domain of CcmE that in the prediction is rotated by 90° towards CcmC. **f** Detail of the CcmCE interface of the AlphaFold2 prediction. The putative heme ligands H60 and H184 are juxtaposed (Fig. 3e), and H130 and Y134 of CcmE that are involved in heme binding are near the heme-binding site.

glyco-diosgenin (GDN), but otherwise both approaches were similar and seem compatible, raising the question which stoichiometry is physiologically correct. We consider it more likely to lose subunits than to gain them, but in the absence of further evidence this is no strong point. However, we unexpectedly also obtained a Ccm(AB)$_2$CD complex when analyzing the E154Q[A] variant that is deficient in ATP hydrolysis and leads to a closed state of the ABC transporter module. The closure of the dimeric transporter induced a conformational change of helices $h$I and $h$V in only one monomer of the CcmB dimer that is not compatible with the observed binding mode of CcmC to CcmB (Fig. 2h), as on the cytoplasmic side of the membrane, helix $h$V of CcmB was pushed outward, leading to a clash with helix $h$IV of CcmC (Fig. 2h). It seems feasible that such a rearrangement results in the effective ejection of the CcmCD module from the complex, and it is quite remarkable[17,37] that it occurs only in one protomer of the CcmB dimer, while the other almost completely retains its open-state conformation with CcmCD in place. Note, however, that all analyses were carried out with solubilized membrane complexes, so that one important aspect of a biological membrane is lost, the lateral pressure of the lipid bilayer. This feature has been discussed most thoroughly in the context of mechanosensitive channels[40], but is a general property that depends on the lipid composition and the shape of the membrane. The loss of lateral membrane pressure in a detergent micelle may well destabilize the CcmABCD complexe, resulting in the dissociation of one CcmCD unit. If so, then what happens in a native environment where the membrane can keep an octameric complex intact? If CcmC does not dissociate, it must accommodate the protruding helix $h$V of CcmB by dislocating its own helix $h$IV (Fig. 2h). This helix is connected to its adjacent helix $h$III on the periplasmic side via the loop containing the WxWD motif, and for the heme cofactor handled by this motif to be passed from the front to the back side of CcmC, as suggested above, the distance between these two helices of CcmC must increase transiently. The rearrangement of Ccm(AB)$_2$ upon closing may provide exactly the driving force required to trigger this sequence of events. This implies the existence of a second conformation of CcmC that remains unobserved to date and that might be difficult to identify due to its transient nature. In this conformation, heme could pass between the two helices and under the WxWD loop, likely sandwiched by the two tryptophan side chains of the latter. In this process the cofactor is rotated, and its charged propionate side chains are turned towards the membrane. This positioning should be unfavorable in the hydrophobic environment of the membrane, but in CcmC the heme in its binding position is already slightly lifted out of the membrane, with one propionate side chain forming a salt bridge to arginine R128. The rotation of the heme group is fundamentally important, as both CcmE in the heme translocase complex and the apopeptide in the lyase complex must attack the vinyl side chains that would be deeply buried within the membrane in the energetically most favorable orientation of the cofactor within the lipid bilayer.

In their recent manuscript, Li et al. describe CcmABCD as a 'heme release complex', implying that the free enthalpy of ATP hydrolysis is used to covalently link the cargo heme to CcmE and eject it from the membrane[19]. This is based on the established finding that even in the absence of CcmAB a CcmCDE complex links heme covalently to the chaperone CcmE[17,37]. However, the so far unobserved, transient conformation of CcmC that we suggest based on the above reasoning would be one where the cofactor can directly interact with the WxWD motif in CcmC. Given the analogy to CcmF and CcsBA, this should occur before the heme group reaches the binding pocket observed in the Ccm and Ccs structures (Supplementary Fig. 5). In this model, ATP hydrolysis serves to change the conformation of CcmC such that its helices $h$III and $h$IV come sufficiently far apart for a heme group to be passed below the WxWD loop and rotated to allow H130[E] to attach to the 3-vinyl group. Both hypotheses are not contradictory: The transport of one heme cofactor to and through the WxWD motif may well be

coordinated with the ejection of another one that already is present in the attachment site from the previous cycle, and ATP hydrolysis then would indeed also eject the cofactor after its attachment to CcmE. The key question is where the heme group resides before it interacts with the WxWD motif of CcmE. Free heme readily partitions into a biological membrane with the charged propionate groups oriented towards the membrane-water interface[18]. Like any amphiphile, the cofactor will have good lateral mobility within the lipid bilayer but will face a high energy barrier for the propionates to cross from the inner to the outer leaflet, so that flip-flop events will be rare. We propose that it is this step where the Ccm(ABCD)$_2$ complex comes into play, acting as a floppase that translocates a heme group from the inner to the outer leaflet and then passes it through the WxWD loop on to CcmE. As the CcmAB knockout studies have shown, this does not seem to be indispensable and an arrested CcmCDE complex is still formed[17,37], but this would only require minuscule amounts of heme present in the outer leaflet, which may well be achieved through other pathways or occur spontaneously. Phylogenetically, CcmAB groups with other ABC transporters that act as floppases, with known structures for the O-antigen transporter Wzm-Wzt[41], and the teichoic acid exporter TarGH[42]. However, we refrain from overstating this analogy, as in the model we present, the cargo would not be transported through the CcmB dimer, but rather along the CcmBC interface. This is not the case for other known floppases, and while we cannot exclude that the CcmB dimer has a so far unseen, outward-open conformation, the sum of our data is better in line with a translocation mode that involves CcmC and its WxWD motif. We further note the analogy to the lipid scramblase nhTMEM16, where lipid transport also occurs along an outer groove of the integral membrane protein that otherwise serves as a chloride channel[43].

Using the energy of ATP hydrolysis to shuttle heme groups from the inner to the outer leaflet of the membrane, Ccm(ABCD)$_2$ thus acts to keep a cellular pool of laterally mobile CcmE chaperones loaded with cofactors, ready to associate with the lyase component CcmFGH(I) for the covalent attachment of its cargo to an apocytochrome binding motif. As such it constitutes an essential part of this complex posttranslational modification machinery, but the question why system II heme maturases of the Ccs system can dispense of this mechanism remains to be answered. Remarkably, they are found in organisms with genomes rich in cytochromes $c$, and the available structural data indicates that these lyases also possess the lateral gate towards the outer leaflet that we observed in CcmF and suggest to be the site of heme insertion by CcmE[18]. At present, central questions remain to be answered, but the picture that increasingly comes into focus for heme maturase assemblies is that of a strikingly mechanical, post-translational modification machinery that consists of independent, moving parts coordinated through conformational changes that are driven by cytoplasmic ATP hydrolysis and cellular redox processes.

## Methods

### Cloning, gene expression and protein purification
The $ccm$ operon of *Escherichia coli* was amplified from the cytochrome $c$ maturation helper plasmid pEC86[21], and cloned into a pASK-IBA vector (IBA Lifesciences), introducing an N-terminal StrepTag(II) (WSHPQFEK) to the gene product of $ccmA$. Recombinant Ccm complexes were produced in *E. coli* strains BL21(DE3) or C43(DE3)[44]. Cultures were grown in TB medium at 37 °C and 180 rpm agitation until the OD$_{600}$ reached 1-2, then protein production was initiated by addition of 0.2 μg·mL$^{-1}$ of anhydrotetracycline and the temperature was reduced to 20 °C. After 14-16 hours of induction, the cultures were harvested by centrifugation at 5000 × $g$ at 4 °C (Avanti J-26 XP, Beckman Coulter). The cell pellets were resuspended in lysis buffer with 50 mM Tris/HCl pH 7.5, 300 mM NaCl, and disrupted by ultrasonication (Branson sonifier 450 W, 70% amplitude, 3 s working and 10 s pause). The crude extracts were then centrifuged at 30,000 × $g$ for

30 min at 4 °C to remove cell debris, and the supernatants were further ultracentrifuged at 300,000 × *g* for 1 h at 4 °C to sediment the membrane fraction. The membrane pellets were resuspended in 8 mL lysis buffer per gram of membranes, solubilized with 1% *n*-dodecyl-β-*D*-maltopyranoside (DDM) for 1 h, and cleared by centrifugation at 100,000 × *g* for 30 min. The supernatants were loaded on a Strep-Tactin Superflow high-capacity column (IBA Lifesciences, 5 ml bed volume) equilibrated in lysis buffer with 0.01% DDM. After washing, the protein complexes were eluted with lysis buffer containing 5 mM D-desthiobiotin, concentrated with ultrafiltration, and further purified by gel filtration using HiLoad Superdex 200 16/600 (GE healthcare) equilibrated in SEC buffer (20 mM Tris/HCl pH 7.5, 150 mM NaCl, 0.01% DDM or 0.02% GDN). Detergent exchange to glyco-diosgenin (GDN) was performed during the affinity purification step. The purified complexes were concentrated and further analyzed using SDS-PAGE. The polypeptide bands on the SDS-PAGE were identified by mass spectrometry (Toplab). All bacterial strains, plasmids an primers are summarized in Supplementary Table 2.

### Cryo-EM grid preparation and data acquisition
The samples of CcmABCDE (GDN) and Ccm($^{E154Q}$AB)$_2$CD in a DDM micelle after gel filtration were concentrated to 10 mg·mL$^{-1}$, applied to glow-discharged Quantifoil R 2/1 grids on 300 copper mesh, incubated for 5 s, blotted for 2 s with filter paper and flash-frozen in liquid ethane cooled by liquid nitrogen using a Vitrobot Mark III (Thermo Fisher) at 90% humidity and a temperature of 10 °C. The dataset for CcmABCDE in GDN was collected using SerialEM[45] on a 200 kV Glacios cryo-transmission electron microscope (Thermo Fisher) equipped with a Gatan K3 direct electron detector at a pixel size of 0.87 Å·px$^{-1}$, 30 frames, and an exposure time of 4 s, with a total dose 50 e$^-$·Å$^{-2}$. The data set for Ccm($^{E154Q}$AB)$_2$CD in DDM was collected on a 300 kV Titan Krios TEM (Thermo Fisher) equipped with a Gatan K2 detector at a pixel size of 0.82 Å·px$^{-1}$, 30 frames, and an exposure time of 4 s, with a total dose 50 e$^-$·Å$^{-2}$.

### Image Processing, model building and refinement
The detailed workflow for data processing is summarized in Supplementary Figs. 3 and 6. For the CcmABCDE (GDN) dataset, the raw movie stacks were motion-corrected with Relion[46], and the per-micrograph defocus values were estimated using CTFFIND v4.1[47]. Initial particle picking was done with a Laplacian-of-Gaussian blob detection and followed by template-based picking in Relion. The extracted particles were then subjected to several rounds of 2D classification, and particles selected as good were used to generate an initial 3D model. After 3D classification, two classes showed well-resolved density for secondary structure features, with an apparent twofold symmetry indicating the Ccm(ABCD)$_2$ assembly. One 3D class also showed additional density for CcmE, and was refined to 3.81 Å resolution after CTF refinement and Bayesian polishing[48] in Relion and non-uniform refinement[49] in CryoSPARC[50]. Particles of these two 3D classes were also combined and posed with C2 symmetry to reach a higher resolution of 3.47 Å for the Ccm(ABCD)$_2$ complex.

The dataset for Ccm$^{E154Q}$ABCD (DDM) was processed using CryoSPARC[50]. Patch motion correction and CTF estimation were done before particle picking using blob picker. The selected 2D classes were used as input for template picker. After the second round of 2D classification, selected 2D classes were used for an ab initio reconstruction. Several rounds of heterogenous and non-uniform refinement improved the final map resolution to 3.94 Å with a composition of Ccm($^{E154Q}$AB)$_2$CD. A second dataset for Ccm$^{E154Q}$ABCD (GDN) with 2.5 mM ATP added was processed analogously, to a final map resolution was 3.62 Å. The gold standard Fourier shell correlation (FSC) 0.143 criterion was used as a map resolution estimate[51] (Supplementary Fig. 6). Cryo-EM maps were visualized using UCSF ChimeraX[52].

A starting model for each subunit of CcmABCDE was generated by SWISS-MODEL[53], and rigid fitted into the density maps using ChimeraX[52]. Model building was performed manually in Coot[54]. After the release of AlphaFold2[24,55], a model of CcmE derived by this algorithm was used to fit the Ccm(ABCD)$_2$E map. For all calculations, a local installation of AlphaFold 2.1 was used on a GPU Server. Improvements of the initial models were carried out with real-space refinement in PHENIX[56], the quality of the structure was validated by MolProbity[57]. Data collection and refinement statistics are summarized in Supplementary Table 1. Figures were generated with PyMOL (Schrödinger LLC) or UCSF ChimeraX[52].

### Reporting summary
Further information on research design is available in the Nature Portfolio Reporting Summary linked to this article.

## Data availability
The density maps and atomic coordinates have been deposited with the Protein Data Bank (PDB) at http://www.pdb.org. The cryo-EM density maps were also deposited with the Electron Microscopy Data Bank (EMDB). Accession numbers are: Ccm(ABCD)$_2$: PDB 8CE1 and EMD-16597, (Ccm(ABCD)$_2$E): PDB 8CE8 and EMD-16601, (Ccm($^{E154Q}$AB)$_2$CD): PDB 8CEA and EMD-16602, and (Ccm($^{E154Q}$AB)$_2$CD with ATP): PDB 8CE5 and EMD-16599.

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

## Acknowledgements
The pEC86 plasmid was a generous gift by Linda Thöny-Meyer. This work was supported by the European Research Council (grant no. 310656) (O.E.) and Deutsche Forschungsgemeinschaft (CRC 1381, project ID 403222702, CRC 992, project ID 192904750, and RTG 2202, project ID 46710898) (O.E.) We thank M. Chami for excellent assistance with data collection at the BioEM lab of Basel Biozentrum. We further acknowledge the bwHPC Cluster of the federal state of Baden-Württemberg and the Deutsche Forschungsgemeinschaft (grant INST 35/134-1 FUGG) for computational support.

## Author contributions
L.I., A.B., L.Z. and O.E. designed the experiments, L.I., L.D., and L.Z. performed the experiments, L.I., L.Z., and O.E. processed data, L.I. and L.Z. built and refined the structural model, and L.I., L.Z., and O.E. wrote the manuscript.

## Funding

## Competing interests
The authors declare no competing interests.
