## [Peer Review File · Nature Communications]

Architecture of the Heme-translocating CcmABCD/E Complex required for Cytochrome c MaturationREVIEWER COMMENTS

Reviewer #1 (Remarks to the Author):

In this manuscript Ilcu, L. et al describe cryo-EM structures of the CcmABCD/E complex that mediates Cytochrome c maturation in E. coli. The authors succeeded in purifying Ccm(ABCD)₂, Ccm(AB)₂CD, and Ccm(ABCD)₂E complexes in various conformational states. In total these structures highlight conformational changes that likely modify the stoichiometry of the overall complex, and provide insight into how ATP binding results in conformational shifts of the core ABC transporter Ccm(AB)₂ and associated subunits. Moreover, computational modeling of these complexes using AlphaFold revealed critical insights into conformations and subunit assemblies that were not observed in the experimental structures. From the experimental cryo-EM structures, AlphaFold predictions, and analysis of previous structural studies, the authors compile a convincing argument for how heme is flipped across the membrane and shuttled through the complex to facilitate cytochrome c maturation. I found the discussion in this manuscript particularly well crafted, and appropriately explained some of the unusual findings and their implications.

For full disclosure I am not an expert in Cytochrome c maturation processes, which involve a highly complicated series of enzymatic reactions and conformational shifts. Rather, I approached review of this paper with a personal background in cryo-electron microscopy and ABC transporter membrane protein complex structure and dynamics. Despite my limited background in cytochrome c maturation processes I found that the authors provided a detailed yet well explained synopsis of their findings on this deeply complicated system. The figures in the manuscript are very well made and helped immensely for me to digest the complicated processes described in this manuscript. Given the recent work on CcmABCD systems by this group as well as Li et al., I feel as though the current manuscript is a logical extension on recently published previous research, and presents new and interesting data that helps to paint a fuller picture of these elaborate membrane protein complexes. In my opinion this manuscript is well composed, timely given similar yet distinct recent results from other groups, and suitable for publication in Nature Communications.

I only have several small critiques, and suggestions that I think would strengthen the paper for a broader audience...

1. Page 6, middle of the page describing residue R54^D. The authors claim this residue “forms three short hydrogen bonds to the short helix at the C-terminus of CcmD”. I believe the sentence should read “forms three short hydrogen bonds to the short helix at the C-terminus of CcmC (not CcmD)?

2. The authors claim that CcmABCD as well as CcmABCDE both co-purified with bound heme b in an oxidized state, as indicated by an α -band at 560 nm upon reduction. It would be good to show this spectroscopic data (maybe additional panel in Suppl. Fig 1). Additionally, it would be good to include the similar data showing lack of this co-purified heme for the E154QA complex.

3. At the extracellular side of the membrane the open and closed states of CcmAB showed minimal differences, and no apparent binding site for a cargo molecule (heme) was identified. In the discussion the authors suggest that “the cargo would not be transported through the CcmB dimer but rather along the CcmBC interface”. Such a transport mechanism seems inconsistent with what has been observed for other ABC transporters that act as floppases by capturing a substrate molecule within the central cavity of the ABC transporter transmembrane domains. It was not clear to me why the authors speculate that the cargo is transported along the CcmBC interface. Is it simply due to a lack of potential translocation pathway identified in the structures? How can the authors rule out the possibility that other unobserved conformational states exist that would allow heme flopping through the central CcmBB interface?

4. Although the figures in the paper are very well constructed, and as a structural biologist I was able to visualize the complex stoichiometries and conformational changes described, I think that a few supplementary videos could go a long way in helping a non-structural oriented audience to comprehend the complicated system described in this manuscript. I would suggest one supplemental video showing the overall stoichiometry of the complex purified under different conditions. A second video could show the conformational changes observed between the Ccm(ABCD)₂ and Ccm(AB)₂CD complexes, and the changes induced by ATP binding to the E154Q^A Ccm(AB)₂CD complex. Finally, a video showing a morph between experimental and AlphaFold structures would be very helpful.

Reviewer #2 (Remarks to the Author):

This is the second cryo em structure of the CcmABCD complex but the first to have some occupancy by CcmE, and the structural consequences of the CcmA E154Q variant There is quite a lot of speculation but that reflects the complexity of this cytochrome c maturation system. There is certainly significance for the field The work appears to be technically of high standard and probably reproducible throughout. It looks like minor differences in handling may affect classes of particles one sees by cryo EM. That is both a potential strength and possible cause of confusion. At this stage it would be unreasonable to expect all labs to spend ages seeking to reproduce others' work by using the same exact constructions, detergents etc

I have only minor points:

1. It is stated that hemes are added sequentially from N- to C terminus. Seems very likely but is there evidence for this? No citation is given.

2. Is it truly established which vinyl group of heme is bound to H130 of CcmE?

3. Could the authors provide a statement as to what vinyl group 2 has become in the newly adopted nomenclature. And similarly for vinyl 4? would help most readers

4. It is almost remarkable that quite a lot was worked out about this system before structural biology could be applied but perhaps the last sentence of the discussion should be At present central questionS...?

Reviewer #3 (Remarks to the Author):

Cytochromes c play central role in electron transfer in respiration and photosynthesis, which require the attachment of heme cofactors by a cytochrome c maturation machinery. The maturation system I found in α - and γ -proteobacteria, archaea, plant and some protozoal mitochondria consists of two integral membrane protein complexes: the heme transporter CcmABCD and the heme lyase CcmFHI. CcmABCD transfers heme cofactors across the membrane and delivers them to the mobile heme chaperone CcmE in the periplasm, which then delivers the cofactor to CcmFHI that allows the covalent attachment of the heme cofactors to apocytochrome. Previously, the authors reported the crystal structure of the core lyase protein CcmF and proposed a mechanism for CcmE in guiding heme cofactors within the lipid bilayer to CcmF. In the current study, Ilcu and colleagues proceeded to investigate the first step of the maturation process to understand how the ABC transporter CcmABCD triggers heme transport to CcmE with expense of ATP. The authors determined cryo-EM structures of Ccm(ABCD)₂ and Ccm(ABCD)₂E complexes at respective 3.47 and 3.81Å resolutions. Although relevant structures of heme-bound Ccm(AB)₂CD have been reported before (Li et al., 2021), the current study showed distinct heterooctameric Ccm(ABCD)₂ configuration and further investigated the interactive details of the periplasmic chaperone protein CcmE to the complex. Comparative analysis of experimentally obtained open or closed and AlphaFold estimated native structures revealed some conformational changes in the helices of CcmB and CcmC and the periplasmic domain of CcmE.

However, apart from the octameric stoichiometry of the Ccm(ABCD)₂ structure determined here, which potentially implicates the symmetric nature of the complex, the authors did not demonstrate adequate

structural distinctiveness of their cryo-EM structures to those reported previously, and their most structural insights are gained from the AlphaFold predictions.

The most remarkable conformational changes observed in the study are in the closed state E154QA structure, in which the helices I and V of one CcmB protomer extend outwards from the cytoplasmic ends (resulted by CcmA dimer closure), which, as the authors discussed, may cause conformational changes of the neighbouring CcmC to induce a “front to back” shift of the heme binding WWD loop. However, unfortunately, the CcmC unit that associated to this CcmB protomer is missing from the E154QA structure (thus the proposed allostery in CcmC is not evidenced), and the other CcmC and CcmB units retained in the complex showed no conformational change. It is paradoxical when the authors reasoned that the release of CcmC was contributed by the conformational changes of CcmB while claiming that CcmB was the allosteric inducer for CcmC. It seems that CcmC is only stable in the complex when CcmB remains the unchanged state. Moreover, how the homodimer of CcmB changes conformations only in one protomer based on the binding of two ATP molecules in the homodimer of the ATPase CcmA is also elusive. The authors have given reasons for these as that the detergent reconstituted proteins used for structural determination is out of the constrain of the native membrane environment, which is plausible. However, this also suggests that the cryo-EM structures obtained here may not reflect physiological state of CcmABCD complex, and lacked structural insights into the ATP-dependent mechanism of heme transfer.

On the other hand, the cryo-EM structure of CcmE-bound CcmABCD revealed novel findings for how CcmE binds to the complex and how the flexible periplasmic domain of CcmE poised for its heme-binding residue H130 to potentially receive heme cofactors from the WWD loop of CcmC, which complement their previous findings of CcmE in context of the other integral complex CcmFHI.

There are some statements in the manuscript need figures to support.

Page 5. 2nd paragraph: CcmB is an integral membrane protein.....situated parallel to the membrane plane and held in place by ionic interactions between the conserved D73 and K96 as well as E10 and R68. (needs to refer to figures)

Page 5. last sentence: The terminal aspartate residue of this motif caps helix IV by accepting short hydrogen bonds from the backbone amide nitrogen atoms of residues A127 and R128.... (Fig. S4). The residue A127 and R128 are not labelled in Fig. S4.

Page 9. Last paragraph: The predicted CcmC subunit....a displacement of the WxWD loop by approximately 1.5 Å this difference was primarily due to a major tilt of the periplasmic half of helix hV of CcmC (Fig. 3e). Such information is not found in Fig. 3e.

Page 10. 1st paragraph: Furthermore, H184 now formed a hydrogen bond from its N atom to the β-carboxylate of D126 of the WxWD motif. (needs to refer to figures).

Fig. 2h: Figures from the two panels seem to be the same.

Fig. 3a: The two CcmB protomers in the figure are labelled as inner and outer, but in the main text they are referred as protomer 1 and protomer 2. It would be clearer to use consistent terms.

It would benefit the general readers if the manuscript could add a schematic diagram for the overview of the cytochrome c maturation system I pathway.

Architecture of the Heme-translocating CcmABCD/E Complex required for Cytochrome *c* Maturation

Lorena Ilcu, Lukas Denkhaus, Anton Brausemann, Lin Zhang & Oliver Einsle

RESPONSE TO REVIEWER COMMENTS

Reviewer #1 (Remarks to the Author):

In this manuscript Ilcu, L. et al describe cryo-EM structures of the CcmABCD/E complex that mediates Cytochrome *c* maturation in *E. coli*. The authors succeeded in purifying Ccm(ABCD)₂, Ccm(AB)₂CD, and Ccm(ABCD)₂E complexes in various conformational states. In total these structures highlight conformational changes that likely modify the stoichiometry of the overall complex, and provide insight into how ATP binding results in conformational shifts of the core ABC transporter Ccm(AB)₂ and associated subunits. Moreover, computational modeling of these complexes using AlphaFold revealed critical insights into conformations and subunit assemblies that were not observed in the experimental structures. From the experimental cryo-EM structures, AlphaFold predictions, and analysis of previous structural studies, the authors compile a convincing argument for how heme is flipped across the membrane and shuttled through the complex to facilitate cytochrome *c* maturation. I found the discussion in this manuscript particularly well crafted, and appropriately explained some of the unusual findings and their implications.

For full disclosure I am not an expert in Cytochrome *c* maturation processes, which involve a highly complicated series of enzymatic reactions and conformational shifts. Rather, I approached review of this paper with a personal background in cryo-electron microscopy and ABC transporter membrane protein complex structure and dynamics. Despite my limited background in cytochrome *c* maturation processes I found that the authors provided a detailed yet well explained synopsis of their findings on this deeply complicated system. The figures in the manuscript are very well made and helped immensely for me to digest the complicated processes described in this manuscript. Given the recent work on CcmABCD systems by this group as well as Li et al., I feel as though the current manuscript is a logical extension on recently published previous research and presents new and interesting data that helps to paint a fuller picture of these elaborate membrane protein complexes. In my opinion this manuscript is well composed, timely given similar yet distinct recent results from other groups, and suitable for publication in *Nature Communications*.

I only have several small critiques, and suggestions that I think would strengthen the paper for a broader audience...

1. Page 6, middle of the page describing residue R54^D. The authors claim this residue “forms three short hydrogen bonds to the short helix at the C-terminus of CcmD”. I believe the sentence should read “forms three short hydrogen bonds to the short helix at the C-terminus of CcmC (not CcmD)?

Correct. We thank the reviewer for spotting this and have corrected it in the revision.

2. The authors claim that CcmABCD as well as CcmABCDE both co-purified with bound heme b in an oxidized state, as indicated by an α -band at 560 nm upon reduction. It would be good to show this spectroscopic data (maybe additional panel in Suppl. Fig 1). Additionally, it would be good to include the similar data showing lack of this co-purified heme for the E154QA complex.

The requested spectra have been included in a revised Extended Data Fig. 1.

3. At the extracellular side of the membrane the open and closed states of CcmAB showed minimal differences, and no apparent binding site for a cargo molecule (heme) was identified. In the discussion the authors suggest that “the cargo would not be transported through the CcmB dimer but rather along the CcmBC interface”. Such a transport mechanism seems inconsistent with what has been observed for other ABC transporters that act as floppases by capturing a substrate molecule within the central cavity of the ABC transporter transmembrane domains. It was not clear to me why the authors speculate that the cargo is transported along the CcmBC interface. Is it simply due to a lack of potential translocation pathway identified in the structures? How can the authors rule out the possibility that other unobserved conformational states exist that would allow heme flopping through the central CcmBB interface?

The reviewer is of course correct that an unobserved, additional conformational state for CcmAB cannot be ruled out. And although most ABC-type floppases group with exporters /type IV acc. to Thomas and Tampé), there are at least two examples for floppases that group into type V together with CcmAB. These are Wzm-WztN and TarGH.

We have added a discussion on this issue to the revised manuscript.

4. Although the figures in the paper are very well constructed, and as a structural biologist I was able to visualize the complex stoichiometries and conformational changes described, I think that a few supplementary videos could go a long way in helping a non-structural oriented audience to comprehend the complicated system described in this manuscript. I would suggest one supplemental video showing the overall stoichiometry of the complex purified under different conditions. A second video could show the conformational changes observed between the Ccm(ABCD)₂ and Ccm(AB)₂CD complexes, and the changes induced by ATP binding to the E154Q^A Ccm(AB)₂CD complex. Finally, a video showing a morph between experimental and AlphaFold structures would be very helpful.

Following the reviewer's suggestion, we have prepared the following animations:

1. *Complex stoichiometries and comparison with the Kranz structure*
2. *Conformational changes upon ATP binding*
3. *Morph with AF structures*

Reviewer #2 (Remarks to the Author):

This is the second cryo em structure of the CcmABCD complex but the first to have some occupancy by CcmE, and the structural consequences of the CcmA E154Q variant There is quite a lot of speculation but that reflects the complexity of this cytochrome c maturation system. There is certainly significance for the field The work appears to be technically of high standard and probably reproducible throughout. It looks like minor differences in handling may affect classes of particles one sees by cryo EM. That is both a potential strength and possible cause of confusion. At this stage it would be unreasonable to expect all labs to spend ages seeking to reproduce others' work by using the same exact constructions, detergents etc

I have only minor points:

1. It is stated that hemes are added sequentially from N- to C terminus. Seems very likely but is there evidence for this? No citation is given.

*Our statement had referred to work from Jörg Simon's lab, who studied cytochromes with different binding motifs that require dedicated heme lyases. However, we realize that this work was primarily done in *W. succinogenes* that employs system II heme lyases for cytochrome c biogenesis. Although all current data point towards very similar modes of function for the lyases of systems I and II, we felt that we should refrain from generalizing here, and we have found no corresponding work for system I. We have therefore removed the statement regarding sequential heme attachment from the revised manuscript.*

2. Is it truly established which vinyl group of heme is bound to H130 of CcmE?

The model of heme binding to CcmE has been revised along the way, as initially a meso carbon had been suggested. The current assumption is that of addition to the 3-vinyl, as reported by the Daldal group whose work we have cited in the manuscript.

3. Could the authors provide a statement as to what vinyl group 2 has become in the newly adopted nomenclature. And similarly for vinyl 4? would help most readers.

The 3- and 8-vinyl groups are denoted in Supplementary Fig. 1a, which follows IUPAC nomenclature. In Fischer nomenclature, the 3-vinyl becomes 2-vinyl and 8-vinyl becomes 4-vinyl. As we could not think of a way to not make this confusing for the reader and considering that the use of IUPAC nomenclature is encouraged we suggest to leave this unaltered.

4. It is almost remarkable that quite a lot was worked out about this system before structural biology could be applied but perhaps the last sentence of the discussion should be At present central questionS...?

Corrected.

Reviewer #3 (Remarks to the Author):

Cytochromes c play central role in electron transfer in respiration and photosynthesis, which require the attachment of heme cofactors by a cytochrome c maturation machinery. The maturation system I found in α - and γ -proteobacteria, archaea, plant and some protozoal mitochondria consists of two integral membrane protein complexes: the heme transporter CcmABCD and the heme lyase CcmFHI. CcmABCD transfers heme cofactors across the membrane and delivers them to the mobile heme chaperone CcmE in the periplasm, which then delivers the cofactor to CcmFHI that allows the covalent attachment of the heme cofactors to apocytochrome. Previously, the authors reported the crystal structure of the core lyase protein CcmF and proposed a mechanism for CcmE in guiding heme cofactors within the lipid bilayer to CcmF. In the current study, Ilcu and colleagues proceeded to investigate the first step of the maturation process to understand how the ABC transporter CcmABCD triggers heme transport to CcmE with expense of ATP. The authors determined cryo-EM structures of Ccm(ABCD)₂ and Ccm(ABCD)₂E complexes at respective 3.47 and 3.81Å resolutions. Although relevant structures of heme-bound Ccm(AB)₂CD have been reported before (Li et al., 2021), the current study showed distinct heterooctameric Ccm(ABCD)₂

configuration and further investigated the interactive details of the periplasmic chaperone protein CcmE to the complex. Comparative analysis of experimentally obtained open or closed and AlphaFold estimated native structures revealed some conformational changes in the helices of CcmB and CcmC and the periplasmic domain of CcmE.

However, apart from the octameric stoichiometry of the Ccm(ABCD)₂ structure determined here, which potentially implicates the symmetric nature of the complex, the authors did not demonstrate adequate structural distinctiveness of their cryo-EM structures to those reported previously, and their most structural insights are gained from the AlphaFold predictions.

We beg to disagree, as the present work shows and discusses the asymmetry in the two CcmB subunits that we consider to be of high functional importance, as well as the essential interaction with the chaperone CcmE. Without this underlying experimental data, the AlphaFold models would arguably be of little value.

The most remarkable conformational changes observed in the study are in the closed state E154QA structure, in which the helices I and V of one CcmB protomer extend outwards from the cytoplasmic ends (resulted by CcmA dimer closure), which, as the authors discussed, may cause conformational changes of the neighbouring CcmC to induce a “front to back” shift of the heme binding WWD loop. However, unfortunately, the CcmC unit that associated to this CcmB protomer is missing from the E154QA structure (thus the proposed allostery in CcmC is not evidenced), and the other CcmC and CcmB units retained in the complex showed no conformational change. It is paradoxical when the authors reasoned that the release of CcmC was contributed by the conformational changes of CcmB while claiming that CcmB was the allosteric inducer for CcmC. It seems that CcmC is only stable in the complex when CcmB remains the unchanged state.

One of the numerous added benefits of studying protein complexes by cryo-EM is that these structures provide a surprisingly good impression of flexibility and dynamics, for instance when combined with 3D variability analyses. On the other hand, it is not a novel concept in enzymology that crucial intermediate states in any process may be high in energy and consequently unstable and sparsely populated in any given sample. Our reasoning here was that the closing motion of the CcmB dimer upon ATP hydrolysis leads to a conformational change in CcmB that is conveyed to CcmC as we describe. It is possible that this other conformation of CcmC is an entatic (strained) state that snaps back faster than structural data is recorded, so that we still have to find a way to trap and visualize it. As the reviewer states a few lines below, we attribute the release of CcmC to the absence of lateral membrane pressure in the detergent-isolated complexes, but we fail to see a paradoxon in this.

Moreover, how the homodimer of CcmB changes conformations only in one protomer based on the binding of two ATP molecules in the homodimer of the ATPase CcmA is also elusive. The authors have given reasons for these as that the detergent reconstituted proteins used for structural determination is out of the constrain of the native membrane environment, which is plausible. However, this also suggests that the cryo-EM structures obtained here may not reflect physiological state of CcmABCD complex, and lacked structural insights into the ATP-dependent mechanism of heme transfer.

P-loop NTPases such as the NBDs of ABC transporters have been studied extensively and in many contexts for decades. Asymmetric nucleotide binding has been observed, as well as an asymmetric reaction to symmetric nucleotide binding. Our reasoning that CcmCD may be ejected upon ATP hydrolysis in detergent-soubilized complexes does explicitly state that we consider the asymmetric complexes to be non-physiological. We do not, however, agree that the data reported by Kranz and co-workers and by us does not provide structural insights into the mechanism of heme transfer.

On the other hand, the cryo-EM structure of CcmE-bound CcmABCD revealed novel findings for how CcmE binds to the complex and how the flexible periplasmic domain of CcmE poised for its heme-binding residue H130 to potentially receive heme cofactors from the WWD loop of CcmC, which complement their previous findings of CcmE in context of the other integral complex CcmFHI.

There are some statements in the manuscript need figures to support.

Page 5. 2nd paragraph: CcmB is an integral membrane protein.....situated parallel to the membrane plane and held in place by ionic interactions between the conserved D73 and K96 as well as E10 and R68. (needs to refer to figures).

The residues have been included into the revised Fig. 1e.

Page 5. last sentence: The terminal aspartate residue of this motif caps helix IV by accepting short hydrogen bonds from the backbone amide nitrogen atoms of residues A127 and R128.... (Fig. S4). The residue A127 and R128 are not labelled in Fig. S4.

The two residues have been omitted, because the H-bonding interaction from residue D26 is only to their backbone amides, so that the side chains do not play a major role. However, we do show this particular arrangement in Fig. 3e, including the labeling of the residues. In this panel, the placement of H184 is from the AlphaFold prediction, but the position of helix hIV and residue D126 is exactly as in the experiment. We have added a reference to Fig. 3d at the corresponding point in the manuscript.

Page 9. Last paragraph: The predicted CcmC subunit....a displacement of the WxWD loop by approximately 1.5 Å this difference was primarily due to a major tilt of the periplasmic half of helix hV of CcmC (Fig. 3e). Such information is not found in Fig. 3e.

We apologize for the mistake. This referred to Fig. 3d and has been corrected.

Page 10. 1st paragraph: Furthermore, H184 now formed a hydrogen bond from its N atom to the β-carboxylate of D126 of the WxWD motif. (needs to refer to figures).

We have added a reference to Fig. 3e, which has exactly this point as its main content.

Fig. 2h: Figures from the two panels seem to be the same.

Panel 2h shows a stereo image set up for wall-eyed viewing, which can be inspected for instance with a viewing aids (e.g. <https://hamptonresearch.com/product-Stereopticon-Stereo-Viewer-438.html>). It provides a three-dimensional impression that helps to understand complex content and is quite commonplace in structural biology.

Fig. 3a: The two CcmB protomers in the figure are labelled as inner and outer, but in the main text they are referred as protomer 1 and protomer 2. It would be clearer to use consistent terms.

We have changed this to consistently use the terms 'inner' and 'outer', as defined in Fig. 3a.

It would benefit the general readers if the manuscript could add a schematic diagram for the overview of the cytochrome c maturation system I pathway.

A schematic of the heme maturation system has been included as a new Extended Data Figure 1 and have re-numbered the subsequent ED figures accordingly.